# HspB4/αA-Crystallin Modulates Neuroinflammation in the Retina via the Stress-Specific Inflammatory Pathways

**DOI:** 10.3390/jcm10112384

**Published:** 2021-05-28

**Authors:** Madhu Nath, Yang Shan, Angela M. Myers, Patrice Elie Fort

**Affiliations:** 1Department of Ophthalmology and Visual Sciences, University of Michigan, Ann Arbor, MI 48105, USA; madhun@med.umich.edu (M.N.); shany@med.umich.edu (Y.S.); prestona@med.umich.edu (A.M.M.); 2Department of Molecular and Integrative Physiology, University of Michigan, Ann Arbor, MI 48105, USA

**Keywords:** αA-crystallin, Müller glial cells, NF-kB, metabolic stress, inflammatory markers

## Abstract

Purpose: We have previously demonstrated that HspB4/αA-crystallin, a molecular chaperone, plays an important intrinsic neuroprotective role during diabetes, by its phosphorylation on residue 148. We also reported that HspB4/αA-crystallin is highly expressed by glial cells. There is a growing interest in the potential causative role of low-grade inflammation in diabetic retinopathy pathophysiology and retinal Müller glial cells’ (MGCs’) participation in the inflammatory response. MGCs indeed play a central role in retinal homeostasis via secreting various cytokines and other mediators. Hence, this study was carried out to delineate and understand the regulatory function of HspB4/αA-crystallin in the inflammatory response associated with metabolic stresses. Methods: Primary MGCs were isolated from knockout HspB4/αA-crystallin mice. These primary cells were then transfected with plasmids encoding either wild-type (WT), phosphomimetic (T148D), or non-phosphorylatable mutants (T148A) of HspB4/αA-crystallin. The cells were exposed to multiple metabolic stresses including serum starvation (SS) or high glucose with TNF-alpha (HG + T) before being further evaluated for the expression of inflammatory markers by qPCR. The total protein expression along with subcellular localization of NF-kB and the NLRP3 component was assessed by Western blot. Results: Elevated levels of IL-6, IL-1β, MCP-1, and IL-18 in SS were significantly diminished in MGCs overexpressing WT and further in T148D as compared to EV. The HG + T-induced increase in these inflammatory markers was also dampened by WT and even more significantly by T148D overexpression, whereas T148A was ineffective in either stress. Further analysis revealed that overexpression of WT or the T148D, also led to a significant reduction of Nlrp3, Asc, and caspase-1 transcript expression in serum-deprived MGCs and nearly abolished the NF-kB induction in HG + T diabetes-like stress. This mechanistic effect was further evaluated at the protein level and confirmed the stress-dependent regulation of NLRP3 and NF-kB by αA-crystallin. Conclusions: The data gathered in this study demonstrate the central regulatory role of HspB4/αA-crystallin and its modulation by phosphorylation on T148 in retinal MGCs. For the first time, this study demonstrates that HspB4/αA-crystallin can dampen the stress-induced expression of pro-inflammatory cytokines through the modulation of multiple key inflammatory pathways, therefore, suggesting its potential as a therapeutic target for the modulation of chronic neuroinflammation.

## 1. Introduction

The unique longitudinal positioning of the Müller glial cells (MGCs) in the retina enables them to interact with every cellular structure of the retina and, hence, makes them a key player in maintaining retinal homeostasis. MGCs are the major glial cell of the retina and as such are involved in multiple processes such as neurotransmitter recycling, preventing glutamate toxicity [1], maintaining the retinoid cycle [2], and regulating nutrient supply [3,4] in a healthy retina. Moreover, MGCs along with pericytes and endothelial cells constitute the blood–retinal barrier and prevent retinal cells from getting exposed to potentially harmful molecules and pathogens [5]. Therefore, any acute or chronic injury to the retina would primarily influence the proper functioning of MGCs and can affect the entire retina’s well-being.

During prolonged or extensive retinal injury such as diabetic retinopathy (DR), MGCs’ stress response is primarily protective through secretion of growth factors and regulation of other pro-survival effects but can eventually become detrimental including through its participation in the inflammatory response with the secretion of various cytokines and other mediators in the extracellular space and the vitreous cavity. Studies have reported that diabetes-induced gliosis eventually leads to the production and release by MGCs of inflammatory molecules such as interleukin-1β (IL-1β), interleukin-6 (IL-6), tumor necrosis factor-α (TNF-α), and chemokine ligand-2 (CCL2) [6,7]. The discovery of these same elevated inflammatory molecules and cytokines in the vitreous of diabetic patients at various stages of DR strongly supports that MGCs play a key role in the synthesis of these cytokines and, more generally, in the neuroinflammatory response associated with DR [8,9,10,11,12].

Lately, there has been a growing interest in the role of crystallin proteins in the glia and especially their implication in neuroinflammation. α-Crystallins have particularly received increasing attention as they were shown to be highly expressed in glial cells in multiple neurodegenerative diseases such as multiple sclerosis [13], Alzheimer’s disease [14], and diabetic retinopathy [15,16]. α-Crystallins are multifunctional proteins that possess remarkable chaperone and anti-apoptotic properties [15,16,17,18]. These proteins are involved in various healthy and stress-associated cellular mechanisms including regulation of protein aggregation and oxidation, through which they can play a central role in inflammation and apoptosis, including those involved in neurodegenerative disorders. Studies have reported that pre-treatment of α-crystallin significantly diminishes the silver nitrate–stimulated acute systemic inflammation in mice models. The pre-treatment of α-crystallin also effectively reduced the glial fibrillary acidic protein (GFAP), nuclear factor kappa B (NFκB), and dopamine, norepinephrine catabolism in the mice neocortex [19,20].

In the retina, an HspB4/αA-crystallin presence was reported in frog [21] as well as mice and rat MGCs [22]. Our group previously reported that HspB4/αA-crystallin was highly expressed in the inner retina of diabetic rats and was primarily located in the ganglion cells and MGCs [15,16]. α-Crystallins undergo numerous post-translation modifications (PTMs), which can affect their chaperone activity [9,23,24,25,26]. Phosphorylation of HspB5/αB-crystallin at Serine residue 59 was shown to be required for multiple functions including its protective role in myocytes [27] or its regulation of lens epithelial cell migration [28]. More recently, phosphorylation of serine residues at positions 19, 45, and 59 of HspB5/αB-crystallin was reported to differentially regulate its chaperone activity for transmembrane proteins [25]. Studies of rodent models of diabetes reported increased phosphorylation of HspB5/αB-crystallin on serine residues 19, 45, and 59 in the retina [15,29,30]. Conversely, we recently reported that phosphorylation on the serine/threonine 148 residue of HspB4/αA-crystallin is high in the retina under normal conditions, while it is dramatically reduced in diabetic donors, especially those with retinopathy [16]. Further exploration revealed that serine/threonine 148 phosphorylation essentially controls the protective role of HspB4/αA-crystallin, highlighting the physiological importance of its downregulation under metabolic stress and diabetic conditions [16]. We have demonstrated that HspB4/αA-crystallin promotes retinal ganglion cell survival under metabolic stress through dampening of the associated endoplasmic reticulum stress. While highly expressed by MGCs, HspB4/αA-crystallin had only a limited effect on ER stress in these cells [16], suggesting a different function of HspB4/αA-crystallin in MGCs. 

The administration of αA in the experimental autoimmune uveitis mice model caused a marked reduction in Th1 cytokines (TNF-α, IL-12, and IFN-γ), both in the retina and in the spleen [31]. Because of the previously demonstrated anti-inflammatory role of HspB4/αA-crystallin in systemic inflammation, we hypothesized that HspB4/αA-crystallin plays a central role in the regulation of the inflammatory response of retinal MGCs during diabetes and metabolic disorders. Therefore, the present study investigated the effect of HspB4/αA-crystallin on the neuroinflammatory response in retinal MGCs under metabolic stress.

## 2. Methods and Materials

### 2.1. Cell Culture

Rat retinal Müller cells (rMC-1) were purchased from Applied Biological Material Inc. (Richmond, BC, Canada). The rMC-1 cell line is derived from the stable transformation of SV40 antigen into primary rat retinal Müller cells shown to express GFAP and cellular retinaldehyde-binding protein (CRALBP), two markers for Müller cells in the adult retina [32]. While originally maintained in 25 mM glucose, cells were progressively acclimated to be cultured and maintained in DMEM containing 5 mM glucose supplemented with 10% FBS. 

Primary Müller glial cells (MGCs) (HspB4−/−) were obtained from the HspB4/αA-crystallin knockout mice originally generously provided by Dr. Wawrousek from the National Eye Institute (NEI). Cells were isolated using a protocol adapted from Hicks and Courtois [33] and characterized as described previously [34]. Briefly, primary MGCs were isolated from the retinal tissue of P10-14 HspB4/αA-crystallin knockout mice pups and maintained in DMEM (5 mM glucose) + 10% FBS + 1% penicillin/streptomycin. The primary Müller cell culture was analyzed for the expression of Müller-cell-specific gene expression from passage 2 to 4.

### 2.2. Transfection and Experimental Protocol

Cells were transfected using the Neon Transfection System (Invitrogen, Waltham, MA, USA) following the manufacturer’s instructions. Briefly, cells were trypsinized and washed in PBS before being resuspended in suspension buffer and electroporated with targeted plasmids. The non-tagged, HspB4/αA-crystallin plasmids used for transfection were generated by point mutation of full-length human sequence in the pcDNA 3.1(+) vector under the control of the human beta actin promotor. All plasmids were subsequently sequence checked. Cells were then plated in six-well plates for gene expression studies. The culture media used in the experiments contained serum unless stated otherwise. At 16 h post-transfection, cells were plated in DMEM with 5 or 25 mM glucose for 24 h. The cells were then incubated in either serum-free DMEM, 25 mM glucose, or 25 mM glucose with 100 ng/mL TNFα (R&D Systems, Catalog # 210-TA, Minneapolis, MN, USA) for four hours before analysis, whereas 5 mM DMEM served as the experimental control. Cells were then harvested for the gene expression studies.

### 2.3. Subcellular Fractionation

Cells were subjected to the REAP (Rapid, Efficient, and Practical) method for subcellular nuclear and cytosolic fractionation as described by Suzuki et al. (2010) [35]. Briefly, the cell pellets were resuspended in 900 μL of ice-cold 0.1% NP40 (Calbiochem, Hayward, CA, USA) in PBS and triturated five times using a p1000 micropipette. Next, 300 μL of the lysate was removed as “whole cell lysate”. The remaining (600 μL) material was centrifuged for 10 s in 1.5 mL micro-centrifuge tubes, and 300 μL of the supernatant was removed as the “cytosolic fraction”. After the remaining supernatant was removed, the pellet was resuspended in 1 mL of ice-cold 0.1% NP40 in PBS and centrifuged as above for 10 s, and the supernatant was discarded. The pellet (~20 μL) was resuspended with 180 μL of ice-cold 0.1% NP40 in PBS and designated as “nuclear fraction”.

### 2.4. Immunoblot

Cells were homogenized by sonication in the previously described RIPA buffer [16]. Protein concentrations were measured with the Pierce BCA reagent, and all samples were adjusted for equal protein concentration. Whole lysates and subcellular fractions were immunoblotted using NuPage gels 4–12% and MES buffer following the manufacturer’s instructions (Thermo Fisher Scientific, Waltham, MA, USA). Gels were run in MES buffer (Thermo Fisher Scientific, Waltham, MA, USA) as per the manufacturer’s instructions. Western blot transfer was carried out on nitrocellulose membranes using the Mini Trans-Blot cell (Catalogue # 1703930, Bio-Rad, Hercules, CA, USA) at 160 V for 1 h at 4 °C. Cell lysates were screened for nuclear factor κB (D14E12, Cell Signaling Technology, Danvers, MA, USA); phosphorylated nuclear factor κB (Ser536) (94H1, Cell Signaling Technology, Danvers, MA, USA); NOD-, LRR-, and pyrin-domain-containing protein 3 (D25PE, Cell Signaling Technology, Danvers, MA, USA); GAPDH (D16H11, Cell Signaling Technology, Danvers, MA, USA); histone H3 (D1H2, Cell Signaling Technology, Danvers, MA, USA); HspB4/αA-crystallin (sc-28306, Santacruz Biotechnology, Dallas, TX, USA) expression; and β-actin (MAB-1501, Millipore, Hayward, CA, USA) as a loading control.

### 2.5. qPCR

The pooled cell lysate samples (*n* = 3) were immediately placed in TRIzol (Thermo Scientific, Waltham, MA, USA), and subsequently, RNA purification was done using the phenol–choloroform method [36]. Isolated RNA samples were quantified by using a NanoDrop 1000 Spectrophotometer (Thermo Fisher Scientific, Waltham, MA, USA). RNA samples of sufficient purity (A260/A280 ratio of 1.9–2.1) were used for the synthesis of cDNA (Thermo Fisher Scientific, Waltham, MA, USA). Contaminating DNA was degraded by treating each sample with RQ1 RNase-free Dnase (Promega, Madison, WI, USA) according to the instruction’s manual. Total RNA isolated from the cell lysates was reverse transcribed into cDNA (Qiagen Omniscript RT kit, 205111, Germantown, MD, USA) as per the manufacturer’s instructions.

The synthesized cDNAs’ amplification was performed using SYBR green master mix (Applied Biosciences A25742, Beverly Hills, CA, USA) and was used for qPCR analysis (Thermocycler, Bio-Rad, Hercules, CA, USA). Real-time experiments were performed in triplicate for each gene including the housekeeping gene (*β-actin*). The threshold cycle at which the increase in the signal with the exponential growth of PCR products was detected was obtained for the quantification. The values were normalized by those in the control group. Relative gene expression analysis was done using the 2-delta delta C(t) method. The primer sequences for the genes tested are listed in Table 1.

### 2.6. Statistics

Gene expression experiment data were normalized to the signal of housekeeping gene *β-actin* and then further with the control group to obtain the delta delta C(t). The mean ± SEM and statistically significant differences are reported. Analyses were performed using non-repeated-measures ANOVA, followed by the Student–Newman–Keuls test for multiple comparisons or two-tailed t-test for a single comparison. A *p*-value less than 0.05 was considered significant.

### 2.7. Study Approval

All experiments were conducted following the Association for Research in Vision and Ophthalmology Resolution on the Care and Use of Laboratory Animals and approved by the Institutional Animal Care and Use Committee of the University of Michigan.

## 3. Results

### 3.1. HspB4/αA-Crystallin Overexpression Dampens the Metabolic-Stress-Induced Pro-inflammatory Transition of Müller Glial Cells (MGCs)

We have previously reported that HspB4/αA-crystallin is highly expressed by retinal MGCs under diabetic conditions. We also showed that rat retinal Müller cells (rMC-1) overexpressing HspB4/αA-crystallin can protect R28 rat retinal neuron cells from metabolic stress [16]. Since, MGCs play a vital role in retinal homeostasis, particularly regarding metabolism, neurotransmission, and inflammation, we further investigated the effect of the HspB4/αA-crystallin expression on the inflammatory response of MGCs to metabolic stress. We observed that metabolic stress induced by serum deprivation leads to elevated expression of pro-inflammatory cytokines including interleukin-6 (IL-6), IL-1beta (IL-1β), and monocyte chemoattractant proteins-1 (MCP-1) in rMC-1 (Figure 1). Interestingly, this serum-deprivation-induced increased expression of pro-inflammatory mediators was significantly prevented by WT HspB4/αA-crystallin overexpression as evidenced by 57%, 80%, and 82% reduction in levels of IL-6, IL-1β, and MCP-1, respectively, as compared to the empty vector. 

Since HspB4/αA-crystallin is induced in MGCs during diabetes, we also studied the effect of HspB4/αA-crystallin overexpression on metabolic stress conditions more reminiscent of diabetes, that is high glucose and TNFα [37,38]. Our data show that exposure of rMC-1 to a “diabetes-like condition” resulted in elevated levels of IL-1β and IL-6 comparable to serum starvation and more than double the effect of TNFα alone (data not shown), demonstrating the synergistic effect of the combination of HG and TNFα compared to either alone. Of note, a much more dramatic increase in MCP-1 expression was observed (352%) in “diabetes-like” conditions when compared to serum starvation. Consistent with a key role of HspB4/αA-crystallin in the regulation of MGC activation in metabolic stress, WT HspB4/αA-crystallin overexpression significantly reduced the induction of IL-6 (61%), IL-1β (77%), and MCP-1 (63%) in the “diabetes-like” condition as compared to the respective empty vector.

### 3.2. HspB4/αA-Crystallin Effect on the Metabolic-Stress-Induced Activation of Müller Glial Cells Is T148 Phosphorylation Dependent

We have previously reported that Thr148 phosphorylation of HspB4/αA-crystallin was reduced dramatically in human donors with diabetic retinopathy. We further demonstrated that the phosphorylation of HspB4/αA-crystallin on residue 148 controls its neuroprotective function in R28 rat retinal neuron cells [16]. Thus, in the same experiment, we also assessed the role of this phosphorylation on the dampening effect of HspB4/αA-crystallin on the metabolic-stress-induced pro-inflammatory response of MGCs. To do so, we overexpressed wild-type HspB4/αA-crystallin (WT), the phosphomimetic (148D) HspB4/αA-crystallin, or the non-phosphorylatable (148A) HspB4/αA-crystallin mutant in rMC-1 cells (Figure 1D). Supporting a key role of this phosphorylation, the phosphomimetic (148D) HspB4/αA-crystallin mutant had an even greater dampening effect than the WT protein on the expression of these pro-inflammatory cytokines, reducing their serum-deprivation induction by 64%, 86%, and 79% for IL-6, IL-1β, and MCP-1, respectively. Conversely, the non-phosphorylatable (148A) HspB4/αA-crystallin mutant was wildly ineffective at reducing the induction of any of these pro-inflammatory cytokines in serum deprivation stress. Similarly, in a “diabetes-like” condition, the phosphomimetic (148D) HspB4/αA-crystallin mutant had an even greater dampening effect than that of the WT HspB4/αA-crystallin as demonstrated by a greater decrease in the induction of IL-6 (78%), IL-1β (79%), and MCP-1 (83%). Conversely again, the levels of induction of IL-6, IL-1β, and MCP-1 caused by “diabetes-like” conditions in presence of the non-phosphorylatable (148A) mutant were comparable to those obtained in the absence of HspB4/αA-crystallin (EV).

### 3.3. Primary Müller Cells Isolated from HspB4/αA-Crystallin Knockout Mice

The data obtained from the rat retinal Müller cell lines (rMC-1) were consistent with a key role of HspB4/αA-crystallin in Müller glial cells, especially in the modulation of their inflammatory response associated with metabolic stress. To further assess the function of αA-crystallin in a more physiologically relevant system, we continued our study in freshly isolated primary MGCs. Additionally, the primary MGCs were isolated from the retinas of HspB4/αA-crystallin knockout mice to prevent any potential interference of the endogenously expressed HspB4/αA-crystallin. The quality and stability of the primary MGCs were controlled by analysis of the expression of specific markers upon isolation. The analysis of expression of genes specific to retinal MGCs, that is *Prdx-6*, *GLUL*, and *Abc8a*, revealed that these cells maintain their specificities in our culture conditions until passage 6 (Figure 2B–D). The dramatic reduction of expression of the specific markers at passage 6 was associated with a change in morphology as the cells became flat and non-polarized (Figure 2A). Henceforth, all experiments were performed with primary MGCs by passage 4.

### 3.4. The Metabolic-Stress-Induced Pro-Inflammatory Response of Primary MGCs (HspB4−/−) Is Dampened by HspB4/αA-Crystallin Expression

Similar to what was observed in rMC-1, serum deprivation also induced increased expression of multiple pro-inflammatory cytokines by primary MGCs (HspB4−/−). Of note, the relative induction of IL-1β, IL-6, and MCP-1 seem even more dramatic in MGCs (HspB4−/−) (≈10–15 fold) as compared to rMC-1 cells (≈5–8 fold), potentially due to the lack of endogenous HspB4/αA-crystallin. Despite this dramatic increase, overexpression of WT HspB4/αA-crystallin in primary MGCs (HspB4−/−) nearly abolished the serum-deprivation induction of these pro-inflammatory cytokines as it resulted in 97%, 88%, 72%, and 89% reduction in expression of IL-6, IL-1β, IL-18, and MCP-1, respectively (Figure 3). Primary MGCs (HspB4−/−) seem to be also very highly reactive to the “diabetes-like” conditions as compared to rMC-1 cells, with even greater induction of the different cytokines analyzed. Under “diabetes-like” conditions, WT HspB4/αA-crystallin overexpression led to the significant reduction of the induction of IL-6 (55%), IL-1β (36%), IL-18 (93%), and MCP-1 (85%) by MGCs (HspB4−/−) (Figure 3).

### 3.5. Phosphorylation of HspB4/αA-Crystallin on Residue 148 Controls the Neuroinflammatory Cascade in Metabolically Stressed Primary MGCs (HspB4−/−)

Here again, we also assessed whether phosphorylation on residue 148 is necessary for the regulation by HspB4/αA-crystallin of the metabolic-stress-induced neuroinflammatory response of primary MGCs (HspB4−/−). Consistent with the key role of this phosphorylation, the metabolic-stress-induced inflammatory response of MGCs (HspB4−/−) was significantly dampened by the phosphomimetic (148D) but not the non-phosphorylatable (148A) HspB4/αA-crystallin mutant. The phosphomimetic (148D) mutant reduced IL-6, IL-1β, IL-18, and MCP-1 expression by 87%, 86%, 64%, and 28%, respectively, while the expression of these inflammatory mediators remained comparable to the empty vector control in MGCs (HspB4−/−) overexpressing the non-phosphorylatable (148A) HspB4/αA-crystallin mutant. Under “diabetes-like” stress, the phosphomimetic (148D) HspB4/αA-crystallin mutant was even more effective at reducing the pro-inflammatory mediator’s response of MGCs (HspB4−/−) than the WT HspB4/αA-crystallin protein. While the phosphomimetic (148D) HspB4/αA-crystallin overexpressing MGCs (HspB4−/−) showed a greater reduction of the “diabetes-like” stress-induced pro-inflammatory mediators evaluated, the non-phosphorylatable (148A) HspB4/αA-crystallin mutant was completely devoid of the dampening effect (Figure 3A–D).

### 3.6. HspB4/αA-Crystallin Regulates MGCs Inflammatory Response through Stress-Specific Inflammatory Pathways

Previous studies have reported that diabetes and metabolic stress can cause the induction of pro-inflammatory cytokines through activation of nuclear factor kappa B (NF-κB) or the NLRP3 inflammasomes [39,40,41]. In addition to demonstrating the key role of HspB4/αA-crystallin in regulating the pro-inflammatory response of MGCs to metabolic stress, our data also demonstrated that it did so despite a cytokine profile varying based on the nature of the stress. This stress specificity of the response prompted us to assess the role of HspB4/αA-crystallin in the expression and activation of these respective pathways. This analysis showed that serum deprivation led to robust activation of the NLRP3 inflammasome (Figure 4A–C) while diabetes-like stress caused a specific and dramatic induction of NF-κB (Figure 4D). While HspB4/αA-crystallin overexpression led to some degree of variability in the expression levels of various effectors of the inflammasome under diabetes-like conditions, none of them were significantly different from normal EV conditions. Conversely, serum deprivation had only a very moderate effect on NF-κB expression compared to the effect of diabetes-like stress. Consistent with its impact on pro-inflammatory cytokines expression, overexpression of WT HspB4/αA-crystallin or the phosphomimetic mutant led to a significant reduction of Nlrp3 (85%), Asc (61%), and caspase-1 (68%) induction in serum-deprived MGCs (Figure 4A–C). We further explored the impact of HspB4/αA-crystallin, and its modification at T148, on the activation of the inflammasome and NF-kB pathways in these respective stresses. Consistent with the effects seen at the transcript level, overexpression of the phosphomimetic mutant significantly prevented the cytosolic Nlrp3 protein induction by serum deprivation (Figure 5A,B). Furthermore, consistent with the transcriptional data, overexpression of WT HspB4/αA-crystallin or the phosphomimetic mutant nearly abolished the “diabetes-like” stress activation of NF-kB. Indeed, overexpression of either one of the WT and T148D mutants resulted in dramatic reduction of NF-kB phosphorylation at Ser536, a well-recognized marker for activation of this pathway (Figure 5C–F). Further underscoring the key role of T148 phosphorylation and consistent with the effects on cytokine expression, overexpression of the non-phosphorylatable (148A) HspB4/αA-crystallin mutant was completely ineffective under either stress condition (Figure 5A–D).

## 4. Discussion

The present study demonstrated the critical role of HspB4/αA-crystallin in the regulation of a metabolic-stress-induced inflammatory cascade in retinal Müller glial cells. This study revealed that HspB4/αA-crystallin overexpression leads to the dampening of the metabolic-stress-induced pro-inflammatory transition of retinal MGCs. It also clearly established that this function is tightly regulated by the phosphorylation of HspB4/αA-crystallin on residue 148, phosphorylation required to suppress the neuroinflammatory cascade in metabolically stressed MGCs. Interestingly, our study also demonstrated the pleiotropic nature of this regulation demonstrated by its efficacy in two different models of metabolic stresses involving partially separate inflammatory pathways. Overall, our study suggests that HspB4/αA-crystallin and its phosphorylation on residue 148 could play a central role in regulating the stress response of retinal MGCs during diabetes and other metabolic disorders.

Our study has demonstrated that retinal MGCs upon exposure to metabolic stress initiate pro-inflammatory cascades. The finding is consistent with previous reports describing the activation of retinal MGCs during neurodegenerative diseases such as diabetic retinopathy [42,43,44,45]. Various studies have reported that increased expression of inflammatory markers such as IL-6 [7], IL-1β [46], and MCP-1 [43] inactivated retinal MGCs due to diabetes-induced stress. This cytokine profile is consistent with the one reported in the current study when primary MGCs are exposed to hyperglycemia and high TNFα, subsequently referred to as “diabetes-like” stress conditions. Altogether, recent studies are strongly supportive of the notion that an inflammatory cascade in activated retinal MGCs is involved in the pathogenesis of diabetic retinopathy [5,47]. Our group has previously demonstrated that HspB4/αA-crystallin is predominantly expressed in ganglion cells and MGCs in diabetic patients [16]. 

While HspB4/αA-crystallin function in retinal MGCs is largely unknown, an active role of HspB4/αA-crystallin in suppressing systemic inflammation has previously been reported [19,20]. Together with the findings of the current study, it supports a regulatory role of HspB4/αA-crystallin in the inflammatory response of MGCs in the stressed retina. Reactive astrogliosis, while being disease and stimulus dependent, is most often associated with a pro-inflammatory A1 phenotype rather than the more protective, anti-inflammatory A2 phenotype. Similarly to inflammation itself, while first necessary and protective, sustained or unresolved reactive gliosis becomes part of the pathophysiology of neurodegeneration, including in diabetic retinopathy. The current study is the first to show the potential role of HspB4/αA-crystallin in regulating retinal MGCs’ activation, including preventing or suppressing the transition to the pro-inflammatory A1 reactive phenotype associated with metabolic stress. 

Post-translational modifications in general, and phosphorylation in particular, are key mechanisms of regulation of protein function, a phenomenon well-characterized for αB-crystallins. Phosphorylation of HspB5/αB-crystallin can be induced by MAPKs including extracellular-signal-regulated kinase (ERK) and p38 under stress conditions [48,49], which eventually regulates its intracellular distribution, translocation, and chaperone activity [50,51] and also enhances its anti-apoptotic potential [52]. Although, earlier in vivo studies had shown HspB4/αA-crystallin in the lens can be phosphorylated at residue-122 and 148, unlike HspB5/αB-crystallin, detailed functional studies for HspB4/αA-crystallin phosphorylation are very limited [53,54,55]. HspB4/αA-crystallin expression was reported to protect astrocytes against C2-ceramide- and staurosporine-induced cell death, and subsequent work suggested that phosphorylation of HspB4/αA-crystallin at residues 122 and 148 enhanced this protective effect [50]. Our group was first to report that Thr148 phosphorylation of HspB4/αA-crystallin was reduced dramatically in the retina of human donors with diabetic retinopathy [16]. While this study also demonstrated that HspB4/αA-crystallin phosphorylation on this specific residue played a central role in regulating its protective function in neurons, it did not unveil the intracellular role of αA-crystallin and its phosphorylation in MGCs. Using primary MGCs, the current study demonstrated the physiological role of HspB4/αA-crystallin in regulating the inflammatory response of those cells when exposed to metabolic-stress conditions. 

Retinal inflammation is a hallmark of various retinal pathologies including diabetic retinopathy, age-related macular degeneration, uveitis, and retinopathy of prematurity. Despite sharing the common feature of retinal inflammation, these diseases vary regarding the pro-inflammatory cytokine profiles associated and the inflammatory pathways involved. This is consistent with our study, as both metabolic stresses tested showed an overall increase in pro-inflammatory cytokine expression, however with different patterns. Most notably, the more chronic “diabetic-like” stress was strikingly associated with a greater increase in MCP-1. Interestingly, this difference was correlated with the marked induction of nuclear factor-kappa B (NF-kB). This observation suggests that MGCs could be directly responsible for the previously reported diabetes-induced elevation of NF-kB, which has been suggested to result in the associated increased levels of MCP-1 [39]. This is further supported by the observation of local expression of MCP-1 and NF-kB in the retina as well as their accumulation in the vitreous of patients suffering from proliferative diabetic retinopathy [56,57]. 

In the more acute metabolic stress induced by serum deprivation, the significant induction in the IL-1β, ΙL-6, and IL-18 was particularly associated with an increase in NLRP3 inflammasome induction. Retinal MGCs are an essential element of retinal innate immunity [58], especially so in response to acute stress that can result in reactive gliosis, including through the induction of IL-1β, ΙL-6, TNF-α, and IL-18 [59,60,61]. The results of the current study also suggest that MGCs can play an important role in regulating the expression of pro-inflammatory cytokines such as IL-1β, ΙL-6, TNF-α, and IL-18 in the early stages of diabetic retinopathy, which was proposed to be associated with the inflammasome activation [62,63]. Recent studies in streptozotocin-induced diabetic rodents showed increased retinal expression of NLRP3, caspase-1, and ASC [46], whereas dysregulation of NLRP3 inflammasome in the Akimba mice led to the continuous activated state of microglia resulting in the loss of the blood–retinal barrier [64]. Studies focusing on age-related macular degeneration have also reported NLRP3-inflammasomes-dependent cytokine induction of IL-1β and IL-18, both locally and systemically [40,41]. Our data suggest that HspB4/αA-crystallin can also dampen the inflammatory mediator induction in the more chronic and “diabetic-like condition” controlled by NF-kB, a key modulator of the expression of cytokines and inflammatory molecules associated with diabetes complications [65,66]. Our data also highlight the role of HspB4/αA-crystallin in the regulation of inflammasome induction, which plays an important role in the more acute inflammatory response of MGCs to metabolic stress-driven insult occurring in the early stages of DR. 

## 5. Conclusions

In conclusion, the data gathered in this study demonstrate the central regulatory role of HspB4/αA-crystallin in MGCs, as it can significantly dampen the detrimental effect of sustained expression of pro-inflammatory cytokines via its differential modulation of major inflammatory pathways. This study points to a novel role of HspB4/αA-crystallin and its modulation by phosphorylation on T148, suggesting its potential use as a therapeutic target for modulation of chronic neuroinflammation. 

## Figures and Tables

**Figure 1 jcm-10-02384-f001:**
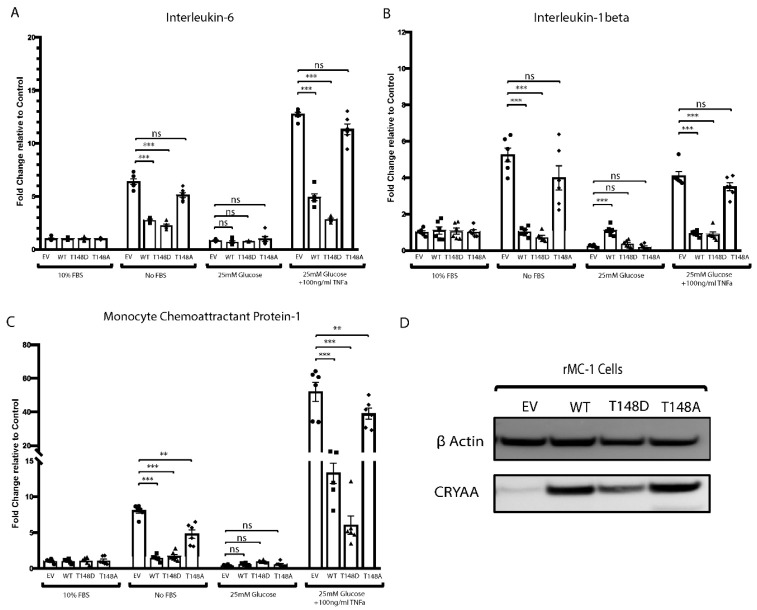
Metabolic-stress-induced expression of pro-inflammatory mediator is regulated by HspB4/αA-crystallin expression. rMC-1 cells were transfected with either empty vector (EV), wild-type HspB4/αA-crystallin (WT), phosphomimetic form of HspB4/αA-crystallin (T148D), or non-phosphorylatable form of HspB4/αA-crystallin (T148A). Then, 24 h post transfection, cells were either exposed to normal media (10% FBS), serum starvation (no FBS), high glucose (25 mM glucose), or to diabetic like stress (25 mM glucose + 100 ng/mL TNFα) for 4 h. The mRNA levels of (**A**) interleukin-1 beta (IL-1β), (**B**) interleukin-6 (IL-6), and (**C**) monocyte chemoattractant protein-1 (MCP-1) were normalized to the actin-encoding gene *Actb*. (**D**) Representative immunoblot depicting the expression of HspB4/αA-crystallin in transfected rMC-1 cells. ** *p* ≤ 0.01, *** *p* ≤ 0.001, significantly different from the respective EV-transfected cells. Each endpoint was measured on a minimum of six technical replicates. Statistical analysis was performed by one-way ANOVA followed by Student–Newman–Keuls test.

**Figure 2 jcm-10-02384-f002:**
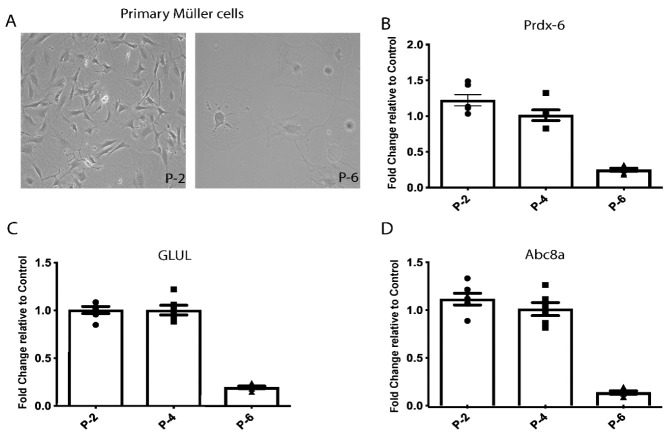
Primary Müller cells isolation from HspB4/αA-crystallin knockout mice retain MGCs’ characteristics. Primary mouse MGCs retain an elongated morphology during the first few passages but become very flat by passage 6 (**A**). The isolated Müller cells show expression of specific markers such as peroxiredoxin-6 (*Prdx-6*), glutamine synthetase (*GLUL*) and ATP-binding cassette 8a (*Abc8a*). Representative graphs of mRNA levels of (**B**) *Prdx-6*, (**C**) *GLUL*, and (**D**) *Abc8a* in primary Müller cells at passage 2 (P-2), passage 4 (P-4), and passage 6 (P-6) are shown normalized to the actin-encoding gene *Actb*. The Müller-cell-specific markers are highly expressed until they dramatically decrease after passage 4.

**Figure 3 jcm-10-02384-f003:**
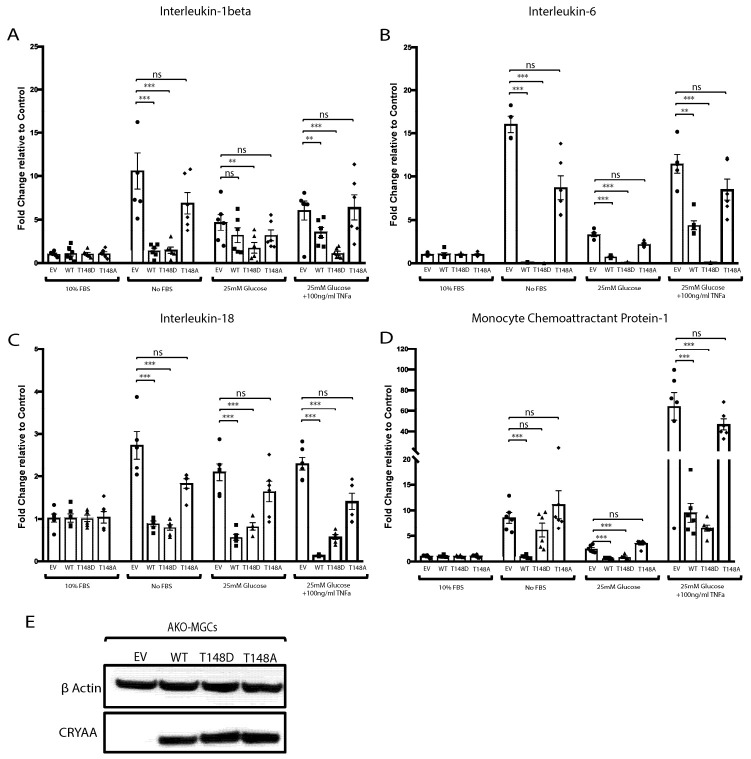
Primary MGCs (HspB4−/−) express pro-inflammatory molecules upon stress exposure, which is modulated by HspB4/αA-crystallin. Primary Müller cells (HspB4−/−) were transfected with either empty vector (EV), wild-type HspB4/αA-crystallin (WT), phosphomimetic form of HspB4/αA-crystallin (T148D), or non-phosphorylatable form of HspB4/αA-crystallin (T148A). Then, 24 h post transfection, cells were either exposed to normal media (10% FBS), serum starvation (no FBS), high glucose (25 mM glucose), or to diabetic-like stress (25 mM glucose + 100 ng/mL TNFα) for 4 h. The mRNA levels of (**A**) interleukin-1 beta, (**B**) interleukin-6, (**C**) interleukin-18, and (**D**) monocyte chemoattractant protein-1 were normalized to the actin-encoding gene *Actb*. (**E**) Representative immunoblot depicting the expression of HspB4/αA-crystallin in transfected MGCs (HspB4−/−). ** *p* ≤ 0.01, *** *p* ≤ 0.001, significantly different from the respective EV-transfected cells. The primary Müller cells (HspB4−/−) were used up to passage 4 for the experiments. Each endpoint was measured on a minimum of six technical replicates in three independent experiments. Statistical analysis was performed by one-way ANOVA followed by Student–Newman–Keuls test.

**Figure 4 jcm-10-02384-f004:**
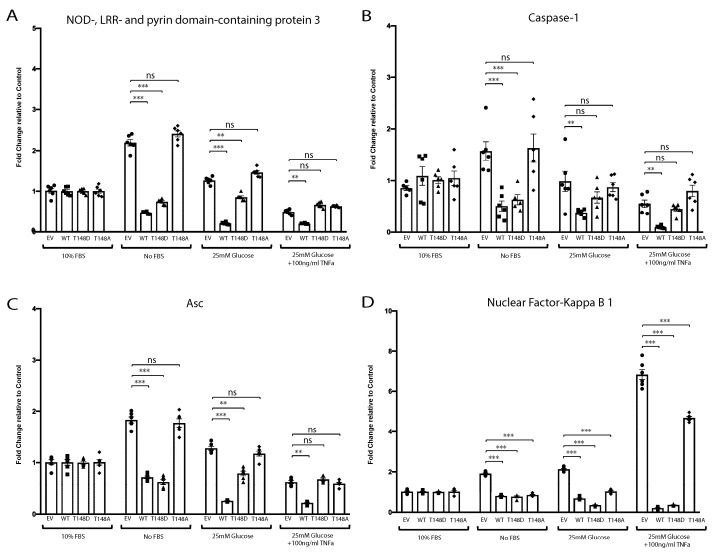
HspB4/αA-crystallin distinct interaction with inflammatory pathways might play key role in regulating neuro-inflammation in MGCs. Primary Müller cells (HspB4−/−) were transfected with either empty vector (EV), wild-type HspB4/αA-crystallin (WT), phosphomimetic form of HspB4/αA-crystallin (T148D), or non-phosphorylatable form of HspB4/αA-crystallin (T148A). Then, 24 h post transfection, cells were either exposed to normal media (10% FBS), serum starvation (no FBS), high glucose (25 mM glucose), or to diabetic like stress (25 mM glucose + 100 ng/mL TNFα) for 4 h. The mRNA levels of (**A**) NOD-, LRR-, and pyrin-domain-containing protein 3 (*Nlrp3*); (**B**) caspase-1 (*Casp-1*); (**C**) apoptosis-associated speck like protein containing a caspase recruitment domain (*Asc*); and (**D**) nuclear factor-kappa B-1 (*NF-kB1*) were normalized to the actin-encoding gene *Actb*. ** *p* ≤ 0.01, *** *p* ≤ 0.001, significantly different from the respective EV-transfected cells. The primary Müller cells (HspB4−/−) were used up to passage 4 for the experiments. Each endpoint was measured on a minimum of six technical replicates. Statistical analysis was performed by one-way ANOVA followed by Student–Newman–Keuls test.

**Figure 5 jcm-10-02384-f005:**
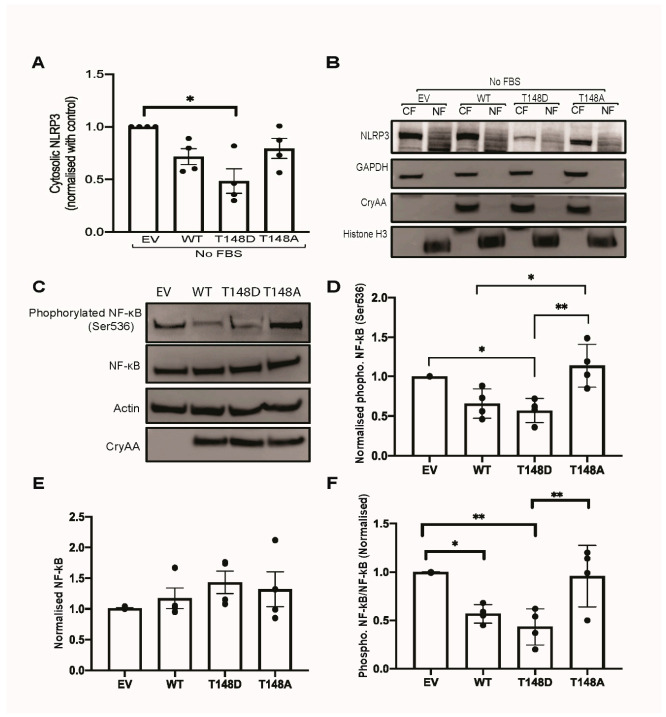
HspB4/αA-crystallin regulates neuro-inflammation in MGCs by inhibiting NLRP3 induction and NF-kB activation. Primary Müller cells (HspB4−/−) were transfected with either empty vector (EV), wild-type HspB4/αA-crystallin (WT), phosphomimetic form of HspB4/αA-crystallin (T148D), or non-phosphorylatable form of HspB4/αA-crystallin (T148A). Then, 24 h post transfection, cells were exposed to either serum starvation (no FBS) or diabetic-like stress (25 mM glucose + 100 ng/mL TNFα) for 4 h. For the inflammasome, NLRP3 subcellular localization was assessed in cytosolic and nuclear fractions and normalized to the empty vector (**A**). Representative images of immunoblot signal for NLRP3, HspB4/αA-crystallin as well as GAPDH and histone-H3, which served as the cytosolic (CF) and nuclear fraction (NF) controls, respectively (**B**). For NF-kB activation, total lysate levels were analyzed for total and phosphorylated NF-kB (Ser536), HspB4/αA-crystallin, and actin, the latter being used as loading control (representative images in (**C**). Fold change of phosphorylated NF-kB (Ser536) (**D**), total NF-kB (**E**), and ratio of phosphorylated NF-kB (Ser536) to total NF-kB (**F**) are shown normalized to the empty vector. * *p* ≤ 0.05, ** *p* ≤ 0.01, significantly different from the respective EV-transfected cells. The primary Müller cells (HspB4−/−) were used up to passage 4 for the experiments. Each endpoint was measured on a minimum of three technical replicates. Statistical analysis was performed by one-way ANOVA followed by Student–Newman–Keuls test.

**Table 1 jcm-10-02384-t001:** Primer sequences of genes analyzed.

*Gene*	Primer Sequence (5′-3′)	
*β-actin*	Forward	GGCTGTATTCCCCTCCATCG
	Reverse	CCAGTTGGTAACAATGCCATGT
*IL-6*	Forward	TAGTCCTTCCTACCCCAATTTCC
	Reverse	TTGGTCCTTAGCCACTCCTTC
*IL-1* *β*	Forward	TTCAGGCAGGCAGTATCACTC
	Reverse	GAAGGTCCACGGGAAAGACAC
*IL-18*	Forward	GACTCTTGCGTCAACTTCAAGG
	Reverse	CAGGCTGTCTTTTGTCAACGA
*CCL2*	Forward	TTAAAAACCTGGATCGGAACCAA
	Reverse	GCATTAGCTTCAGATTTACGGGT
*VEGF*	Forward	GGCCTCCGAAACCATGAACTT
	Reverse	TGGGACCACTTGGCATGGTG
*ICAM1*	Forward	GAGCCAATTTCTCATGCCGC
	Reverse	GCTGGAAGATCGAAAGTCCG
*TNFa*	Forward	CAGGCGGTGCCTATGTCTC
	Reverse	CGATCACCCCGAAGTTCAGTAG
*Nf-kB1*	Forward	TCCACTGTCTGCCTCTCTCGTC
	Reverse	GCCTTCAATAGGTCCTTCCTGC
*ASC*	Forward	CAGCAACACTCCGGTCAG
	Reverse	AGCTGGCTTTTCGTATATTGTG
*Casp1*	Forward	GGAAGCAATTTATCAACTCAGTG
	Reverse	GCCTTGTCCATAGCAGTAATG
*Nlrp3*	Forward	ATTACCCGCCCGAGAAAGG
	Reverse	TCGCAGCAAAGATCCACACAG
*Prdx6*	Forward	CGCCAGAGTTTGCCAAGAG
	Reverse	TCCGTGGGTGTTTCACCATTG
*GLUL*	Forward	TGAACAAAGGCATCAAGCAAATG
	Reverse	CAGTCCAGGGTACGGGTCTT
*Abc8a*	Forward	CGTGGGCCTTATTGTGCAAGA
	Reverse	CAGGTCCACATCAGGCAGTG
*Rpe65*	Forward	ACCACTAACAGCTCATGTCACA
	Reverse	ACAGGTGATAGAAAGGCTCAGAT

## Data Availability

Data generated are included within the manuscript.

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
