# Peer review of "HspB4/αA-Crystallin Modulates Neuroinflammation in the Retina via the Stress-Specific Inflammatory Pathways"

_jcm, 2021, doi:10.3390/jcm10112384_

Round 1

Reviewer 1 Report

The work by Nath et al. aims to investigate the regulatory role of alpha crystallin on MGC activation in serum deprivation conditions or during a “diabetes-like” stress.

The rationale of the study is clear; the study is well designed, and the manuscript is well written. The group have extensive experience and a strong expertise in the field of diabetic retinopathy and glial cells. The proposed study builds on the team’s previous work and clearly identify alpha crystallin as a key regulator of MGC activation. The authors elegantly demonstrate a role of alpha crystallin phosphorylation in the modulation of its activity.

Major comment:

The authors extensively use a condition described as “a diabetic-related stress” where MGCs are stimulated by high glucose (25 mM) and TNFa. It is unclear how this dual stimulation differs from a simple TNF stimulation in normo glucose condition. In line with that the main finding of the study point at a NFKB specific response to the dual stimulation that can be seen as a TNF-only response of MGCs. The authors should provide a control of TNF in normo glucose condition as an additional control to demonstrate that both high glucose and TNF are necessary to reach a diabetic-related stress.

Minor comments:
- it is unclear if the 25mM glucose and the diabetes-like conditions are in the presence or not of FBS. This should be clearly indicated in the figure legends and methods.
- qPCR labelling is not correct, the 2 delta delta ct are not based on log. Please adapt Y axis labelling.

-please provide references for line 208 : “on metabolic stress conditions more reminiscent of diabetes, that is high glucose and TNFα “

- Figure 2 clearly indicates that P6 primary muller cells downregulate the expression of MGC markers while the authors indicate in line 247: “maintain their specificities in our culture conditions until passage 6 (Figure 2B-D)”. please adapt your conclusions. Please indicate if cells were used at P4 or earlier as the authors do not have evidences that Muller cells can be used after P4.

- Authors claim that FBS deprivation results in a NLRP specific activation line 308-309 while diabetes-like conditions caused a specific induction of NFKB. However, their data in figure 4 indicate that FBS or diabetic-related stress results in a very similar 2x increased in NLRP and NKBP expression levels. please adapt your description of the results and conclusion.

Author Response

We want to thank the reviewers and the editor for their very important comments that helped clarify and improve the manuscript. A detailed point by point response to the comments raised by each reviewer is given below.

Reviewer 1:

Point 1: The authors extensively use a condition described as “a diabetic-related stress” where MGCs are stimulated by high glucose (25 mM) and TNFa. It is unclear how this dual stimulation differs from a simple TNF stimulation in normo glucose condition. In line with that the main finding of the study point at a NFKB specific response to the dual stimulation that can be seen as a TNF-only response of MGCs. The authors should provide a control of TNF in normo glucose condition as an additional control to demonstrate that both high glucose and TNF are necessary to reach a diabetic-related stress.

Response 1: While the field of DR has gathered a lot of evidence relative to a potential role of HG in many cells (including some in Müller cell lines), the goal of this study was to assess how HG alone differ from its effect in conjunction with TNFa, since TNFa has been demonstrated as one of the major pro-inflammatory cytokines elevated in animal models and patients with DR. Thus, this study explored the impact of this accumulation and its potential interaction with the hyperglycemic milieu of diabetes. In the present study, we thus elected to compare HG alone to HG+TNFa and demonstrated that HG alone only has minimal effect on MGCs glial activation but in combination with TNFa, it leads to a synergistic induction of the MGCs activation that is greatly dampened by aA-crystallin in a T148 dependent manner. We agree with the reviewer and acknowledge that TNFa by itself does induce some degree of response as demonstrated in the literature as well as in our optimization experiments on EV transfected cells. Indeed, in these preliminary experiments we tested TNFa alone and observed a 2 fold increase in NFKB expression at the same time point (4h; data not shown), which is far less than the more than 6 fold increase reported here with the combination of high glucose (HG: 25mM) and TNFa (100ng/mL; Figure 4), demonstrating the synergistic effect. A sentence has been added to the manuscript to clarify this point.

Point 2: it is unclear if the 25mM glucose and the diabetes-like conditions are in the presence or not of FBS. This should be clearly indicated in the figure legends and methods.

Response 2: Thank you for the suggestion. Changes have been made in the method section to clarify this point.

Point 3: qPCR labelling is not correct, the 2 delta delta ct are not based on log. Please adapt Y axis labelling.

Response 3: Thank you for the suggestion. The graphs have been modified and the Y axis re-labelled to reflect fold changes.

Point 4: please provide references for line 208: “on metabolic stress conditions more reminiscent of diabetes, that is high glucose and TNFα “

Response 4: Thank you for the suggestion, appropriate references are now provided.

Point 5: Figure 2 clearly indicates that P6 primary muller cells downregulate the expression of MGC markers while the authors indicate in line 247: “maintain their specificities in our culture conditions until passage 6 (Figure 2B-D)”. please adapt your conclusions. Please indicate if cells were used at P4 or earlier as the authors do not have evidences that Muller cells can be used after P4.

Response 5: We apologize for the lack of clarity, the manuscript has been corrected to reflect that cells were always used by P4.

Point 6: Authors claim that FBS deprivation results in a NLRP specific activation line 308-309 while diabetes-like conditions caused a specific induction of NFKB. However, their data in figure 4 indicate that FBS or diabetic-related stress results in a very similar 2x increased in NLRP and NKBP expression levels. please adapt your description of the results and conclusion.

Response 6: We respectfully disagree with the reviewer on this point. While there is a ~2 fold induction of NLRP3, Caspase-1 and Asc in serum deprivation, there is absolutely no induction of either one of the players of the NLRP3 inflammasome in this stress condition. Conversely, whereas NFKB induction was more than 6 fold in HG+TNFα, a less than 2 fold increase was observed in serum deprivation. 

Reviewer 2 Report

In the present manuscript the authors investigate the role of aA-crystallin/HspB4 in retinal Müller cells. They show that aA-crystallin/HspB4 overexpression reduces secretion of pro-inflammatory cytokines in a Müller glial cell line and in primary cultures of Müller cells of aA-crystallin/HspB4 deficient mice. Furthermore, they provide evidence that this protective effect is mediated by phosphorylation at T148. The data are interesting and provide significant new information in this specific research field. Although the anti-inflammatory effect of aA-crystallin/HspB4 has already been described before, the finding of regulation by phosphorylation at T148 is new and an important information

I have some major concerns regarding the description of the experiments, presentation of data, nomenclature used and references cited.

  • The nomenclature used is outdated. Alpha-crystallins have been shown long time ago to belong to the small heat shock protein family and , thus, have been renamed to HspB4 (aA-crystallin) and HspB5 (aB-crystallin). This makes sense since they share many properties with all the other hspBs and are not related to beta- and gamma-crystallins. Please use the term HspB4 for aA-crystallin or both names HspB4/aA-crystallin. Related to this issue it is misleading or sometimes even wrong to talk about „a-crystallins“ (see introduction line 59 to 80) since aA-crystallin and aB-crystallin are different proteins with different functions, especially in non lenticular tissue. For example reference 13 and 14 refer only to HspB5 and not to HspB4
  • Page 2 , line 71: reference 19 and 20 are cited wrong. These papers do not provide evidence if the described effect was mediated by HspB4 or by hspB5 since a mixture of both proteins were applied in these studies.
  • Page 2 line78 : Reference 27 and 28 show the importance of phosphorylation of hspB5 at S59 and not at S19 and 45!
  • Line 82: In reference 29 phosphorylation was not investigated!
  • Please check all references carefully for correctness. This is not the responsibility of a reviewer!
  • Methods: There are no information given which plasmids were used for overexpression of HspB4 and how the mutants (T148D and T148A) were generated. Which vector was used? Were the proteins tagged?
  • Experiments of figure 1: How was the transfection efficiency controlled? The results provide only valid data if all cultures display comparable transfection rates.
  • Experiments in paragraph 3.1 and 3.2 displayed in figure 1: Were these experiments performed in parallel or after each other? As described in the text one gets the impression that first in one experimental series overexpression of wildtype was performed and in a second set of experiments the phosphorylation mimicking mutants. If this ist the case, results cannot be combined in one figure (no comparison possible). If not, please specify and describe experiments/results correctly.
  • Results 3.4 and 3.5: Please specify that all experiments were performed with MGCs of HspB4 deficient mice and replace MGC with MGC(HspB4-/-).
  • Why does the western blot show a faint band for HspB4 in the knock-out mice? (3E)
  • Again, were experiments performed in parallel (see above)?
  • Figure 4 should be presented before figure 5 in the manuscript.
  • The results and description of experiments of figure 5 are incomprehensible. Please describe in more detail. The abbreviation p65 (D-F) is not explained.
  • Please check references in the discussion for correctness. For example reference 44 is not cited correctly (line 396).
